# Experiments on the Efficacy of Drawing for Memorization among Adults and Children with Varying Written Word Memory Capacities: A Two-Way Crossover Design

Unnur Guðrún Óttarsdóttir

The Reykjavik Academy, 105 Reykjavik, Iceland; unnuro@akademia.is; Tel.: +354-8670277

**Abstract:** In recent years, the comparative effectiveness of drawing and writing for memory has been investigated, but the findings have mostly been analyzed for the entire sample of participants rather than subgroups. In quantitative two-way crossover experiments involving 134 children and 262 adults, drawing for memorization as compared to writing was investigated. The participants were divided into subgroups based on their ability to recall the greatest, moderate, or fewest words and drawings. The difference in the number of recalled words and drawings was then compared between subgroups with varying memory capacities for written words and drawings. Participants who had difficulty remembering written words recalled more drawings than written words relative to participants who remembered written words more easily—this applied to shorter- and longer-term memory. To determine the applicability of the findings to various contexts, the following conditions were varied in four separate experiments: participant age, duration of encoding and recall, number of words memorized, interval between encoding and recall, and the research setting. Drawing benefited memory more than writing in all tested scenarios for the subgroup that remembered the fewest number of words. The new finding of the study is that people who have difficulty remembering written words benefit the most from drawing for memorization compared to those who remember written words more easily and this applies to the various tested conditions.

**Keywords:** drawing; memory; art therapy; memory drawing; education; art educational therapy

## 1. Introduction

Since the turn of the century, several structured studies have been conducted on drawing versus writing for memory, and the results have shown that drawing is generally more effective than writing for memorization [1–15]. Few studies have investigated who benefits the most from drawing for memory [4,5]. In order to study for whom drawing for memorization is most effective in comparison to writing, participants in the present study were divided into subgroups based on the greatest, moderate, or fewest number of recalled written words and drawings. The capacity for remembering through drawing versus writing was compared for the subgroups. Drawing for memory has been shown to be effective under various conditions, including diverse settings, different encoding times, and different numbers of words to be memorized [3]. In the present study, various conditions were applied to investigate whether the subgroup-related research findings would be applicable to various conditions in terms of age, duration of encoding, number of words memorized, intervals between encoding and recall, and different settings.

### 1.1. Memory of Mental Images

Drawing for memory includes preparation, actively constructing and producing an image by drawing, receiving an image by looking at the drawn picture, and visualizing the drawn image in the mind when retrieving it. For a long time it has been known that visualization of mental images aids memory [16–19]. The ancient Greeks created a

memorization technique that applied visualizing images as a memory aid. This method originated when a Greek poet, Simonides, was invited to a party to read poetry. When Simonides went to the door during the party, the house collapsed, killing everyone within. When the relatives of the deceased sought to know which severely injured body was which, Simonides was asked to help identify the victims. He visualized each person in his mind in the setting they had been in during the party, enabling him to recognize every individual [16]. The method of loci, which involves visualizing what one wants to remember in association with a familiar place, originates from this event.

Research that includes therapeutic aims has been conducted in relation to memorizing self-affirming autobiographical memories through the method of loci [20]. One group of participants applied the loci method for the memorization and recall of autobiographical memories, whereas the control group crunched the memories into sets and rehearsed them. There was no significant difference in the amount of recall between the groups one week after the first recall. However, a test conducted one week later showed better recall for the loci group method. This study integrated the method of loci, originally aimed at remembering facts, with a therapeutic aim. The findings of Dalgleish et al. [20] are in line with those of the present study in terms of images facilitating long-term memory to a greater extent than short-term memory. In addition, the findings of the study are in accordance with the application of drawn images when working with memories as part of a therapeutic aim, as in art educational therapy (AET). The quantitative memory drawing study introduced in this article was partly created because of having observed the effectiveness of drawing in a qualitative AET study [7].

### 1.2. Picture Superiority Effect

Although interest in images as a memory aid has existed for a long time, research regarding the memory of pictures was rare until Roger Shepard [21] published a study in 1967 in which participants were shown pictures and words, which they then had to recognize again along with other stimuli not seen before. The participants were 98% accurate in recognizing the pictures but 90% correct when recognizing words. This 'picture superiority effect' has been confirmed in various additional and more recent research studies [22–30].

### 1.3. Drawing for Memory

In 1973, Paivio and Csapo [31] found that drawing the content of words was more effective for memorization than for writing. Art educational therapy (AET), in which coursework learning, including drawing for memorization, is integrated into art therapy, was designed, studied, and developed through a qualitative case study conducted on five children who were dealing with specific learning difficulties and had experienced stress and/or trauma. Indications that drawing facilitated memory of coursework were found in the case studies. To further investigate the memory drawing function of AET, quantitative research was conducted with a large group of participants, that is of 134 children, with the aim of investigating the effectiveness of drawing in comparison to writing in facilitating shorter- and longer-term memory. Descriptive statistics showed that drawing is more effective than writing for memorization in the long term [7]. Statistical analysis for the findings of the study is presented for the first time in this article. In recent years, various studies have confirmed the effectiveness of drawing for memory in relation to different factors and conditions [1–15]. For example, Wammes et al. [3] published findings from a study that compared the memory of written words and the drawn content of words in adults, which showed the effectiveness of drawing in improving memory. Even preparation for drawing for one or two seconds, excluding the actual drawing, has been shown to be more effective for memorization than the act of writing for 15 s [11]. However, Ottarsdottir's [7] research remains the only study that has investigated the memory of drawing in comparison to writing over such a long duration of time as nine weeks, and which showed that as time passes, an

increased number of drawings relative to words are remembered. The study is also the only such structured research comparing the memorization of drawing and writing in which children have participated.

*1.4. Reasons for the Effectiveness of Drawing for Memorization*

Wammes et al. suggested that drawing is effective for memory because it creates a cohesive memory trace that integrates visual, motor, and semantic codes into one memory trace [3] and that it creates vivid contextual information that facilitates later recall [2]. Fernandes et al. [1] proposed that the reason for this is context-rich representation created by integrated distinct codes. Roberts and Wammes [9] suggested that drawing creates a link with novel multisensory information which makes the method effective for memorization. Although these authors mentioned the semantic, sensory, novel, and vivid contextual information, as well as the context-rich representation embedded in drawing, as a reason for its effectiveness for memorization, they did not define precisely what information they were referring to. They did not mention the art therapeutic perspective, which includes an awareness of the personal and emotional material that can be implicit in the drawing. In addition, they did not discuss ethical concerns related to sensitive emotional material that can emerge through memory drawing [8]. From an art therapeutic perspective, drawing is more effective for memorization than verbalized writing, partly because drawing can facilitate connections to deeper, more meaningful personal experiences and emotions, which can in some cases be unconscious (e.g., [32]). From that point of view, the drawn content is stored for longer in memory compared to written words, partly because of its valuable personal and emotional content for the drawer [7].

In order to explore the way in which different types of encoding facilitate memory, Wammes et al. [10] studied the elaborative, motoric, and pictorial components of drawing when memorizing through several encoding techniques. A comparison was made between memorizing through drawing, blind drawing, tracing, imagining, viewing, and writing. The findings showed that the effectiveness of different encoding methods was in the same order, with drawing being the most effective and writing the least effective. Based on the findings of this study, Wammes et al. [10] claimed that integrating multiple distinct sources of information into one cohesive task is beneficial to memory. It is also possible to interpret these findings in the context of the visual non-verbal system on the one hand and the verbal system on the other, where all encoding methods except writing are non-verbal, and all of these are more effective for memorization than verbal writing. The visual modalities are closer to the meaning of the content to be memorized in comparison to verbal writing, as there are more steps from content to writing, where each letter has a sound and then a few letters are combined into a word, which is the label for the content. Possibly, this verbalization translation takes up a lot of the memorization function, while the other non-verbal systems provide the content's meaning without those additional translation steps, and thus, more capacity is available for remembering through non-verbal systems. In addition, in terms of non-verbal drawing methods, drawing and drawing blindly are more effective than tracing, as found in the study conducted by the researchers [10]. The two methods of drawing and drawing blindly both rely on personal creation, while tracing does not, which supports the claim that the effectiveness of drawing for memory is partly due to personal and emotional material within the creation of drawing, even though it is done blindly. Although imagining, which is less effective for memorization than tracing, also has some quality of personal creation, the motoric factor of tracing, in addition to the visual concrete aspects, may override the personal creation included in visualizing, resulting in tracing being more effective for memorization than imagining.

Drawing and paraphrasing for memorization were compared in a study showing that the two methods were comparable [12]. The researchers claimed that both methods apply self-generated elaboration, in which participants develop personal representations.

This again supports the claim that the reason drawing is such an effective memorization technique is partly because of creative, personal involvement in its application.

Tran [15] found that while drawing is more effective for memorization in comparison to writing words down, this difference is greater for positive and negative words than for neutral words. These findings support the claim that the positive and/or negative emotional content of drawing, which according to art therapy theories can be more prominent within symbolic drawing than verbalized written words, is partly the reason for the effectiveness of drawing in comparison to writing.

Additional research has shown that doodling facilitates memory. In a study conducted by Andrade [33], 40 participants were divided into two groups, both of which listened to telephone voice messages where the names of people coming to a party were mentioned. Half of the group wrote down names of people who would be coming to the party while listening to the message, while the other part of the group doodled by shading printed shapes while listening. The participants did not know that afterwards they would be asked to recall the names they heard while they doodled. The research showed that doodling increased recall by as much as 29%, although it was unrelated to the memorized material.

However, Meade et al. [6] found through their research that when participants were asked to either free-form doodle, draw a picture, or write down items to remember, they showed poorer free recall for words encoded during free-form doodling in comparison with words that were drawn or written. In other words, they showed that task-relevant drawing was more successful for memorizing than writing and when applying unrelated doodling.

Andrade [33] and Meade et al. [6] compared somewhat different conditions, as the participants in Andrade's research doodled printed shapes while the comparison group wrote down names while listening, whereas the participants in the research performed by Meade et al. actively doodled, drew, or wrote. What is of particular interest in relation to the present study, apart from unrelated doodling facilitating memory, as shown by Andrade [33], is that the link between the content of the word to be memorized and the drawing created is an important factor in facilitating memory more so than unrelated doodling, as shown by Meade et al. [6]. In AET, various ways of drawing are integrated for educational and emotional purposes [7,8]. One important function is art-making in the context of coursework content. This link between content and drawing is seen as an important component that causes drawing to be effective in terms of coursework learning and memory. Doodling and unrelated spontaneous drawing are also seen as important aspects of AET, both for therapeutic and educational purposes, although they may not result in as much educational achievement as task-related drawing.

*1.5. Application of Drawing for Memory in Education and Therapy*

Various studies have investigated the application of drawing for coursework learning in schools. The methods applied in these studies vary and the findings are mixed [34]. However, as stated previously recent studies have isolated the factor of memorization for drawing in comparison to writing, and the results of those studies are in agreement about drawing being a better memorization technique than writing [1–14].

Wammes et al. [12] compared drawing and note-taking in relation to memorization of definitions from textbooks, and they found indications that drawing was more beneficial than writing down information verbatim. Roberts and Wammes [9] compared memorization through writing and drawing for concrete and abstract words, and found that drawing was generally more effective than writing for both abstract and concrete words, although the memory benefit was greater for concrete words. Jonker et al. [35] found that drawing is effective for remembering individual items, but in order to remember a sequence within a list, it appeared to be more effective to read silently. Jalava et al. [13] investigated, in an educational setting, encoding through drawing based on definition in comparison to copying the definition and they found that drawing was more effective for memorization when recalling both after one and three weeks.

Learning, including memorizing and processing emotions, occurs simultaneously through the drawing process in AET [7]. From an art therapy perspective, the emotional content of the drawing requires an awareness of the person's emotions and the situation in which the drawing is made. Certain security is important for people to feel safe enough to engage with their emotions when drawing for memorization. The educators' or therapists' knowledge of the foundations of art therapy theories and methods is claimed to be important in this aspect to create safety for students who learn and memorize through drawing [8].

### 1.6. Memory Drawing for Specific Subgroups

Most studies that have investigated drawing in comparison to writing for memory have analyzed findings on the number of recalled drawings and written words for the whole sample of participants (e.g., [3,7]), rather than investigating patterns for specific subgroups in order to study who benefits to the greatest extent from drawing for memory. While the results show that the majority of participants recall a greater number of drawings than written words, this is not the case for every participant, as some recall equal numbers of drawings and words, and a few recall more written words than drawings. For example, Wammes et al. [3] mentioned that in two out of seven of their experiments, 26 out of 30 participants in one experiment and 43 out of 49 participants in another experiment recalled a larger number of drawings than written words. However, exploration of the findings in relation to the four and six participants who recalled more written words than drawings was excluded from the discussion of the findings.

A few studies have investigated which population benefits the most from drawing for memory. A comparison of younger and older adults who memorized words and drawings was made by Meade et al. [4] who found that older adults benefited more from drawing for memory than did younger adults. Meade et al. [5] found that drawing enhanced memory in both healthy older adults and people with probable dementia. Although both groups benefited from drawing, they found no difference in the proportional benefit from drawing relative to writing for these two groups. They hypothesized that this may have been because some of the participants had severe dementia symptoms, and thus, the brain regions related to visual perceptual processing had been affected by the disease.

The findings of Meade et al. [4], which show that older adults benefit more from drawing for memorization than younger adults, point to the importance of investigating the effect of drawing on memorization, especially for those who have difficulty remembering written words, as is the case for older people. To further study the effect of drawing on specific groups, the participants in the present study were divided into subgroups consisting of participants who recalled the greatest, moderate, or fewest number of written words on one hand and drawings on the other. The difference between the number of recalled words and drawings was then compared between subgroups. The following research question was posed: Is there a difference in the effectiveness of drawing for memory in comparison to writing for children and adults who remember the greatest, moderate, or fewest words and drawings?

### 1.7. Drawing for Memory within Various Conditions

A study of different time periods, from encoding to recall, showed that the memorization benefit of drawing increased as the duration of time decreased [7]. Drawing was found to be even more effective than writing when the time for encoding was reduced and a greater number of words were encoded. In addition, drawing was found to be effective when testing took place in an individual testing room and in a group classroom setting [3].

To further investigate whether the findings for the subgroups of participants who recalled the greatest, moderate, or fewest numbers of written words and drawings would apply to a variety of conditions, four experiments were conducted under several conditions that differed in the duration of encoding and recall, the number of words memorized, the

time from encoding to recall, age groups, and the setting where the study took place. The research question was as follows:

How many drawn words, in comparison to written ones, are memorized by the whole group of participants and by the subgroups of participants who recalled the greatest, moderate, or fewest number of written words and drawings? Does the difference in capacity to remember through drawing and writing apply across different conditions?

## 2. Materials and Methods

### 2.1. Research Background

The research originated when I worked as an art therapist in a high school in Iceland with adolescents who had specific learning difficulties, where the integration of art therapy and literacy lessons took place. I also worked with children in a public school in Iceland simultaneously as an art therapist and a special education teacher. I combined these two professions as the children drew in relation to their coursework learning and simultaneously worked with their emotional difficulties.

This initial observation of the effectiveness of integrating coursework learning into art therapy led me to conduct qualitative case study research with children who had experienced stress and/or trauma and had specific learning difficulties [7]. The qualitative study inspired me to conduct quantitative research with children, specifically on memorization through drawing in comparison to writing. Data from an additional quantitative study were collected from adults, and the comparison of memory for written words and drawings was investigated under various conditions.

### 2.2. Research Design and Ethics

The quantitative research conducted on the children by the researcher using a two-way crossover design was reviewed and approved by the Icelandic Data Protection Authority. In order to ensure the ethical conduct of the research, the children were free to participate in the research, and no sensitive personal information was collected. Adults who participated in the experiments provided written informed consent for their data to be used for research purposes, which is an ethical requirement requested according to the regulations of the Icelandic Data Protection Authority.

### 2.3. Procedure and Participants

#### 2.3.1. Experiment 1

A sample of 134 children aged 9 to 14 years participated in Experiment 1, which comprised Tests 1, 2, 3a, and 3b. The selection of children took place by including children in randomly selected art education classes, which are compulsory in Icelandic secondary and elementary schools. The random selection of children in art education classes was made by the school authorities prior to the onset of the research. Half of the children in each grade were in one of the art education classes, and those children participated in the research.

The first step in Test 1 involved encoding by drawing the content of 15 words presented in a list of written words on a sheet of paper (Figure 1) (for approximately half of the participants, the first 15 words were encoded by writing them down). This was followed by recalling the words on another sheet of paper. The next step was encoding, which involved writing down a list of 15 words that was presented on yet another sheet of paper (the group that wrote in the first step now drew those 15 words). This was followed by recalling the words through writing them on an additional sheet of paper. Ten minutes were provided to memorize each set of 15 words, which means that the children had an average of 40 s to memorize each word. A detailed description of the study procedures has been reviewed in a previous open-access publication [7].

The results of Test 1 did not show a difference in the median number of recalled written words and drawings immediately after encoding. Drawing may relate to personal conscious and unconscious meanings and memories, which may facilitate storage at a deeper level, which, in turn, might cause the drawing to be stored in memory longer than writing. These speculations resulted in the decision to investigate again, in Tests 2, 3a, and 3b, how much the children recalled after a certain time.

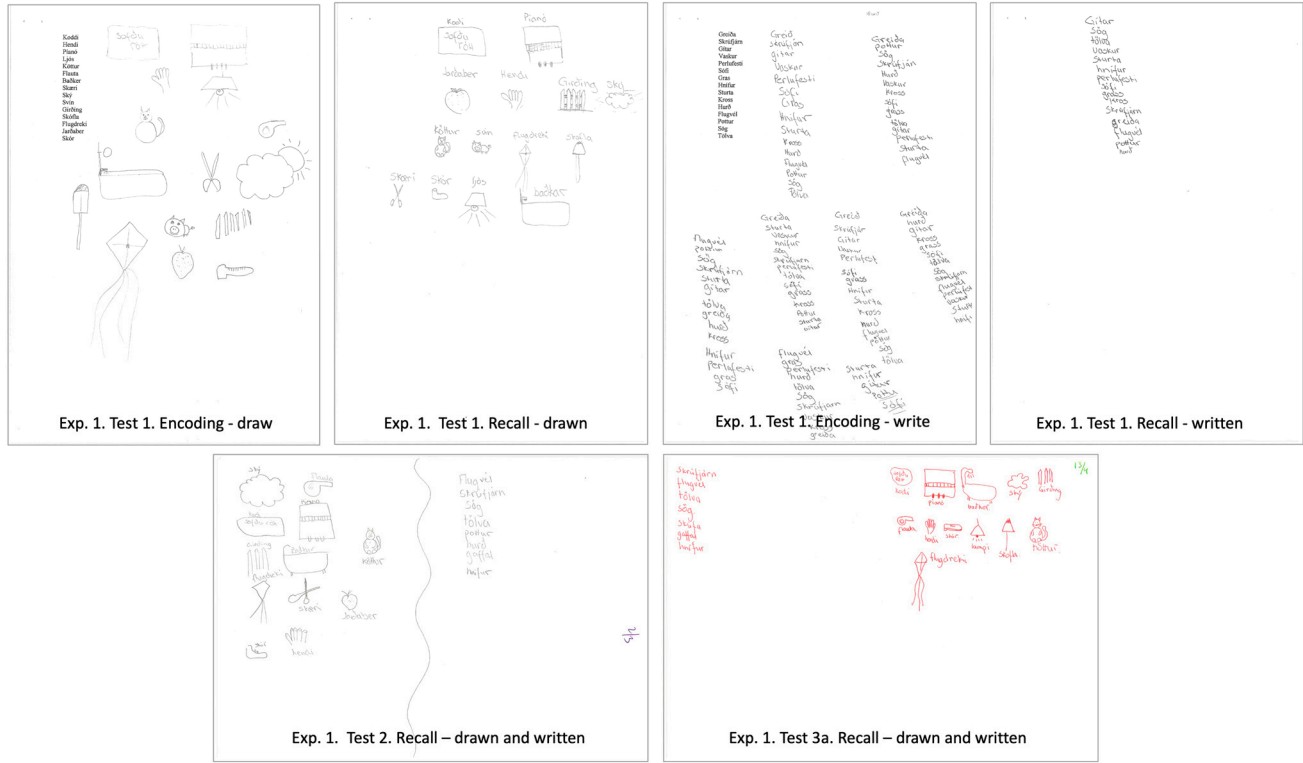

**Figure 1.** Samples from a participant in Experiment 1.

All participants in Experiment 1: Tests 2, 3a, and 3b participated in Test 1. For Test 2, 114 of the 134 children recalled three weeks after encoding in Test 1. For Test 3a, 100 of the 134 children recalled nine weeks after the original encoding, having undergone a recalling and reminding process six weeks earlier when completing Test 2. For Test 3b, 19 children who were absent in Test 2 for various reasons, such as illness, recalled nine weeks after the original encoding without a reminder experience during the nine weeks. The children recalled drawings and written words on one sheet of paper, and they were given unlimited time for the recall process.

### 2.3.2. Experiments 2–4

Experiments 2–4 were conducted with adults to investigate whether the findings of Experiment 1 would apply across different conditions, that is in terms of different age groups, different numbers of words encoded, different durations of time devoted to encoding, various durations of time between encoding and recalling, and different settings. Experiments 2–4 were integrated into lectures given by the researcher on memory drawing research, where the audience was invited to take part in the memory drawing research. The participants completed a memory drawing exercise similar to that in Experiment 1.

The participants in Experiments 2 and 3 who participated in the lectures received a printed research package, while the participants in Experiment 4 who participated in webinars received a digital research package document by email, which they printed, worked with, scanned, or photographed, and then sent to the researcher through email. Instructions on how to conduct the memory drawing exercise were given orally by the

researcher and were also presented in written form on the screen before the list of words to be memorized appeared (Figure 2).

Several numbered sets of words were displayed on a screen. Participants sitting next to each other were allocated different numbers at the beginning of the lecture, and they were instructed to memorize the set of words on the screen that had the same number. Hence, participants sitting next to each other memorized different sets of words.

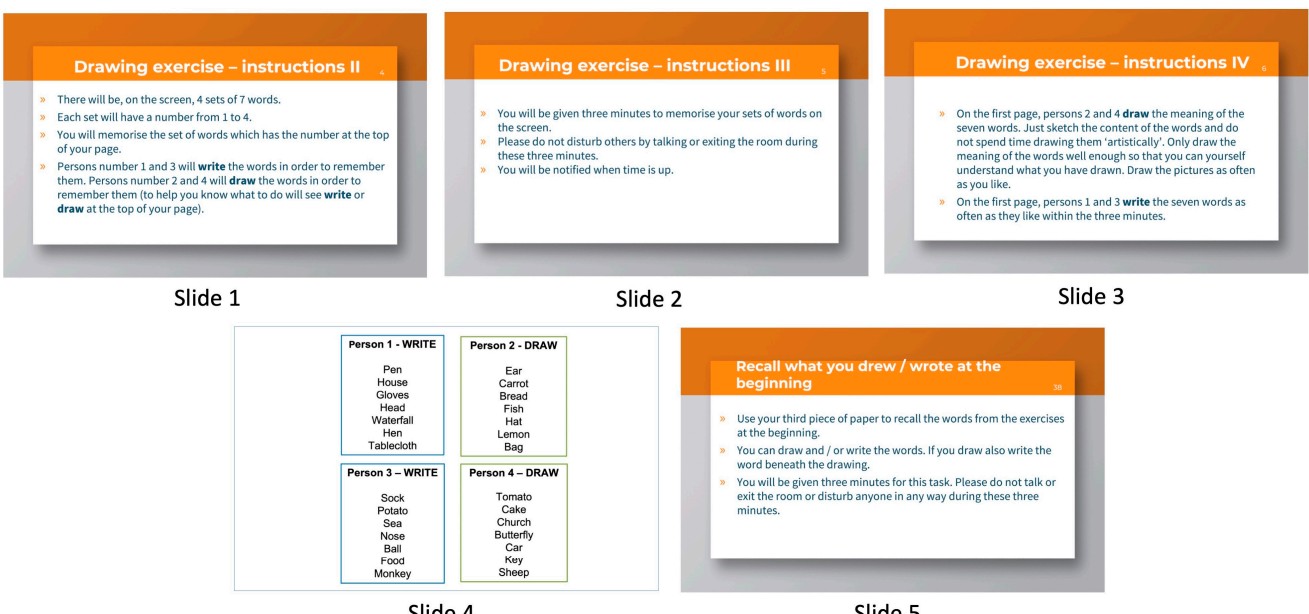

**Figure 2.** Instructions for experiments 2–4. Time and number of words varied between experiments. The instructions (slides 1–3) were repeated, changing "drawing" to "writing" when the participants changed from the drawing to writing step and vice versa.

Each participant was given a four-page research package. The first page was for encoding by drawing the content of the words presented on a screen (for approximately half of the participants, the first words were encoded by writing them down). The second page was used for encoding by writing down other words presented on a screen (the group that wrote in the first step drew the words in this step). The third page was for recalling the words encoded on pages one and two. When some time had passed since the encoding (45 min in Experiment 2 and 4 or 25 min in Experiment 3), the participants recalled what they could remember by drawing and/or writing it on the third page (Figures 3 and 4). When the time was up in each step of the investigation, the participants were told to turn over the page they had been working on and place what they had drawn/written underneath the other pages in the research package. On the fourth page, the participants were invited to submit the findings of their memory drawing exercises and give consent for the anonymous use of the data for research purposes.

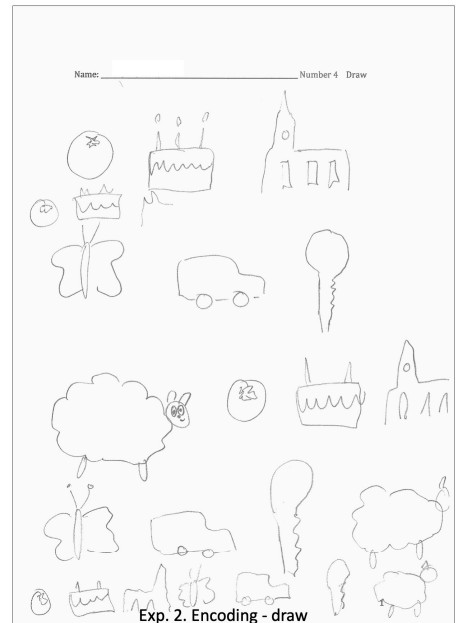
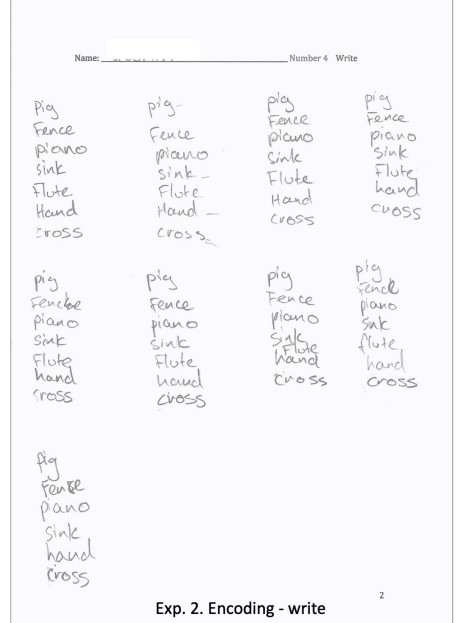
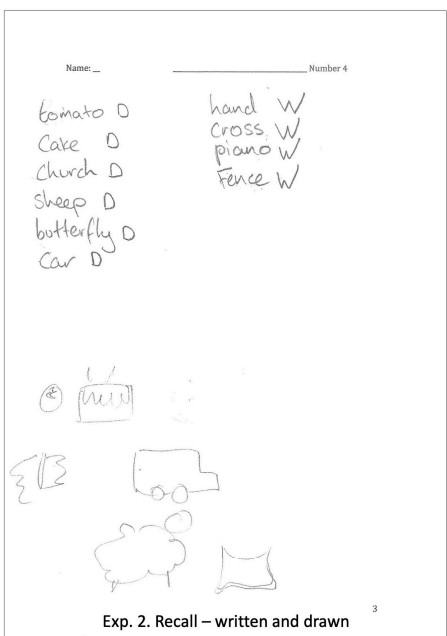

**Figure 3.** Samples from a participant in Experiment 2.

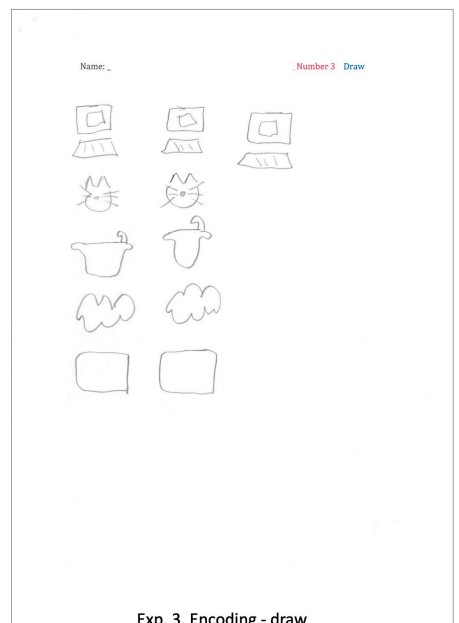
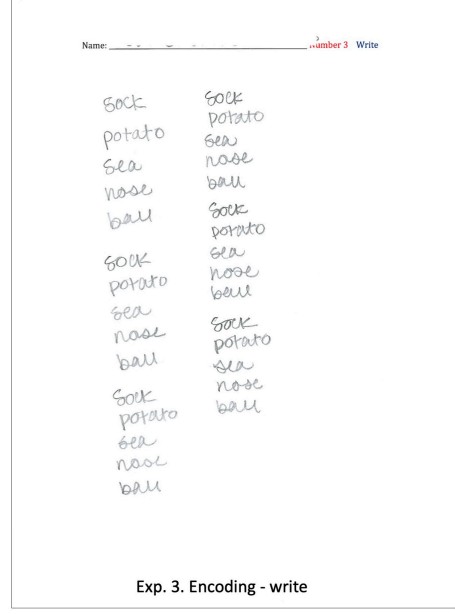
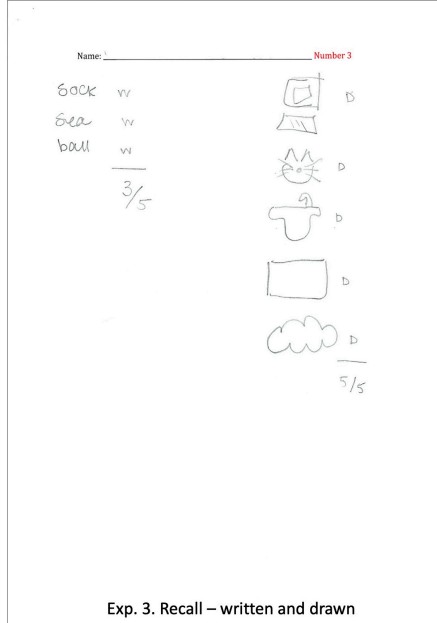

**Figure 4.** Samples from a participant in Experiment 3.

### 2.3.3. Experiment 2

The data for Experiment 2 were obtained from three lectures. First, a keynote lecture that was given at the 20th Nordic Art Therapies Conference held in Iceland, attended by approximately 75 people. Second, four identical public lectures given in Icelandic at the Reykjavik Academy, which were open and free of charge, attended by approximately 80 people. Third, an open lecture at Goldsmiths, University of London, attended by approximately 85 participants. The content, structure, and time length were the same for all lectures.

At the beginning of the lectures, participants were presented with a list of seven words on a screen to memorize by drawing (writing), and then another list of seven words to be memorized by writing (drawing) (Figures 2 and 3). Three minutes were given to memorize

each set of seven words, which means 26 s on average per word. Participants recalled the words approximately 45 min after encoding and were given 3 min for recall. In total, 148 valid responses were collected.

### 2.3.4. Experiment 3

The data for Experiment 3 were obtained from three lectures. First, a lecture that was given at the American Art Therapy Association Conference in Kansas City, attended by approximately 50 people; second, a lecture given at the International Art Therapy Practice/Research Conference in London, attended by approximately 10 people; and third, a lecture given as part of the international Erasmus project 'Social Inclusion and Well-being through the Arts and Interdisciplinary Practices' (SWAIP), attended by approximately 25 people. The content, structure, and length of time were the same for all lectures.

At the beginning of the lectures, participants were presented with a list of five words on a screen to memorize by drawing (writing) and then another list of five words to memorize by writing (drawing). One minute was given to memorize each set of five words, or an average of 12 s per word. Participants recalled the words approximately 25 min after encoding and were given one minute for recall (Figure 4). In total, 47 valid responses were collected.

### 2.3.5. Experiment 4

Data for this part of the research were collected through four online webinars that were free and open to anyone interested. The content and length of the webinars were identical to those of the lectures in which the data for Experiment 2 were collected. The researcher recorded the webinars beforehand and was physically absent during the webinars, including when the participants completed the memory drawing test. A total of 139 participants finished watching the webinar and 67 valid responses were collected.

### *2.4. Validity*

In each experiment, several different lists of words were provided so that participants who sat next to each other encoded and recalled different lists of words. This prevented the participants from seeing the words they were recalling on the adjacent person's sheet of paper. Half the group of participants first memorized by drawing, and the other half memorized by writing. Thus, the order of encoding—either first by drawing or first by writing—did not affect the outcome.

### *2.5. Data Analysis*

Paired sample *t*-tests were used to examine differences in the average number of recalled drawings and written words. To examine whether the extent of the average differences was different for individuals who remembered the greatest, moderate, or fewest written words, participants in each experiment were divided into six equally large subgroups using a tertile split. The groups comprised participants who recalled the greatest (Group 1), moderate (Group 2), and fewest (Group 3) numbers of written words, and participants who recalled the greatest (Group 4), moderate (Group 5), and fewest (Group 6) numbers of drawings. If the group sizes were not exactly equal, the remaining individuals were randomly assigned to one of the subgroups. One-way ANOVA tests were then conducted to test the differences in the effectiveness of drawing relative to writing between the subgroups.

## 3. Results

### *3.1. Number of Words and Drawings Recalled*

For Experiment 1: Test 1, in which the participants were children and recall occurred immediately after encoding, the average number of words recalled was 12 drawings and 12.3 written words (Table 1 and Figure 5). A paired sample t-test did not reveal a significant difference between the average number of recalled written words and recalled drawn

words for children immediately after encoding in Test 1 ($t(133) = -1.16$, $p = 0.248$). The average number of words recalled in Test 2, three weeks after the encoding, was 5.6 drawn words and 2.6 written words. The children who participated in Test 2 recalled significantly more drawings than written words ($t(113) = 10.92$, $p < 0.001$). Nine weeks after the original encoding, the average number of words recalled in Test 3a was 5.3 drawn words and 2.4 written words. The children who participated in Test 3a had a reminding experience six weeks earlier when they completed Test 2. The average number of words encoded in Test 3b, 9 weeks after the original encoding without a recalling experience in between Tests 1 and 3b, was 5.4 drawn words and 1.1 written words. On average, participants recalled significantly more drawn words than written words nine weeks after encoding in both Test 3a ($t(99) = 9.84$, $p < 0.001$) and Test 3b ($t(18) = 6.23$, $p < 0.001$).

**Table 1.** Average, median and percentage of recalled drawings and written words for all experiments and tests. Conditions and variables included in all experiments and tests.

| | Exp. 1 Test 1 | Exp. 1 Test 2 | Exp. 1 Test 3a | Exp. 1 Test 3b | Exp. 2 | Exp. 3 | Exp. 4 |
|---|---|---|---|---|---|---|---|
| Number of participants (*N*) | 134 | 114 | 100 | 19 | 148 | 47 | 67 |
| Age | 9–14 years | 9–14 years | 9–14 years | 9–14 years | adults | adults | adults |
| Number of words encoded | 2 × 15 | 2 × 15 | 2 × 15 | 2 × 15 | 2 × 7 | 2 × 5 | 2 × 7 |
| Average number of drawings recalled (SD) | 12.0 (2.84) | 5.6 (2.69) | 5.3 (2.85) | 5.4 (3.11) | 6.1 (1.04) | 4.3 (0.93) | 6.5 (0.84) |
| Average number of written words recalled (SD) | 12.3 (3.09) | 2.6 (2.53) | 2.4 (2.29) | 1.1 (1.41) | 5.4 (1.83) | 3.5 (1.38) | 5.9 (1.65) |
| Median number of drawings recalled | 13 | 5 | 5 | 5 | 6 | 5 | 7 |
| Median number of written words recalled | 13 | 2 | 2 | 1 | 6 | 4 | 6 |
| Percentage of drawn words recalled out of number of words encoded (%) | 80 | 37 | 35 | 36 | 88 | 86 | 92 |
| Percentage of written words recalled out of number of words encoded (%) | 82 | 17 | 16 | 7 | 77 | 70 | 84 |
| Average time to encode each word | 40 s | 40 s | 40 s | 40 s | 26 s | 12 s | 26 s |
| Time from encoding to recall | immediately after | 3 weeks | 9 weeks | 9 weeks | 45 min | 25 min | 45 min |
| Recalling experience in-between | no | no | after 3 weeks | no | no | no | no |
| Setting | classroom | classroom | classroom | classroom | lectures | lectures | webinars |

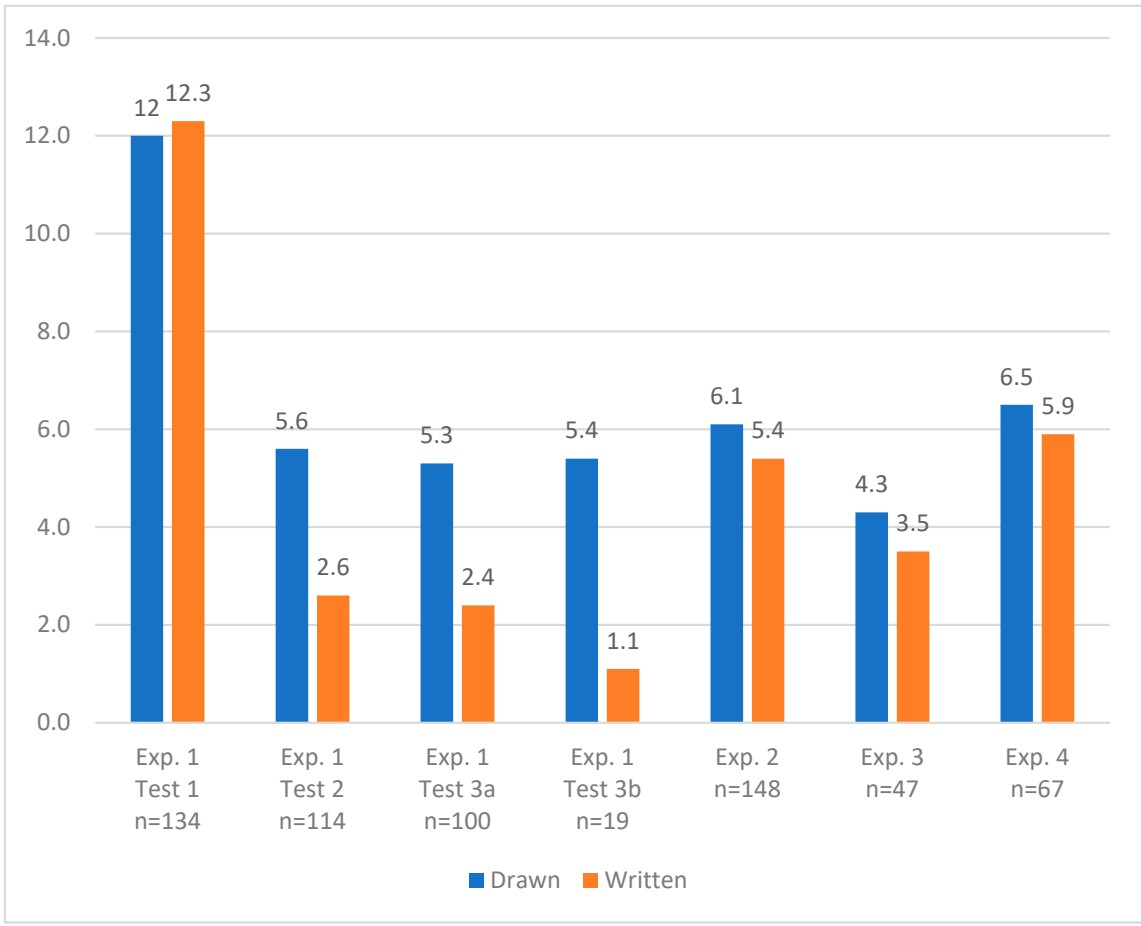

**Figure 5.** Average number of recalled words and drawings. Experiment 1: Test 1, recall immediately after encoding. Experiment 1: Test 2, recall three weeks after Test 1. Experiment 1: Test 3a, recall nine weeks after Test 1, with recall in Test 2. Experiment 1: Test 3b, recall nine weeks after Test 1, without recall in Test 2. Experiment 2, recall 45 min after encoding. Experiment 3, recall 25 min after encoding. Experiment 4, recall 45 min after encoding.

The average number of words recalled in Experiment 2 was 6.1 drawn words and 5.4 written words out of the 7 words encoded 45 min earlier. The average number of words recalled in Experiment 3 was 4.3 drawn words and 3.5 written words out of 5 words encoded 25 min earlier. The average number of words recalled in Experiment 4 was 6.5 drawn words and 5.9 written words out of 7 words encoded 45 min earlier. On average, adult participants recalled significantly more drawn words than written words in all three experiments—Experiment 2 ($t(147) = 5.40$, $p < 0.001$), Experiment 3 ($t(46) = 4.15$, $p < 0.001$), and Experiment 4 ($t(66) = 2.79$, $p = 0.007$).

In summary, paired sample t-tests revealed that participants recalled a significantly greater number of drawings than written words in all experiments and tests except for Experiment 1: Test 1, when the children recalled immediately after encoding. The proportion of written and drawn words recalled out of the number of words originally encoded in all experiments and tests can be found in Figure 6.

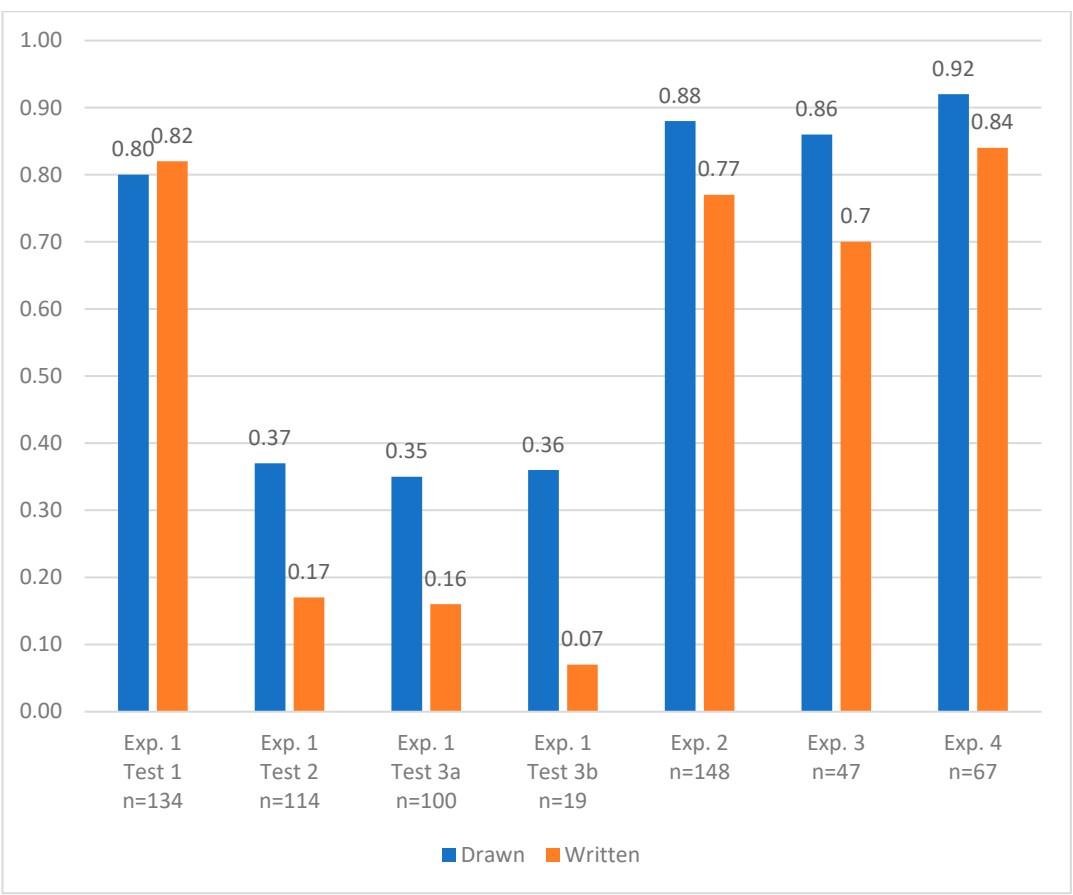

**Figure 6.** Proportion of written and drawn words recalled out of number of words originally encoded in all experiments and tests.

*3.2. Recalled Written Words and Drawings for Specific Subgroups*

To study which groups of children and adults benefited most from drawing in comparison to writing down words for memorization, an additional analysis was conducted in which all participants in each experiment and test were divided into Groups 1–6. Participants who recalled the greatest number of written words were in Group 1, those who recalled a moderate number of written words were in Group 2, and those who recalled the fewest number of written words were in Group 3. Group 4 consisted of participants who recalled the greatest number of drawings, Group 5 consisted of participants who recalled a moderate number of drawings, and Group 6 consisted of individuals who recalled the fewest number of drawings.

The findings of the experiments and tests for Group 3, who recalled the fewest written words, showed that the average number of recalled drawings was significantly greater than that for recalled written words, both for adults and children, in all experiments and tests (Figure 7). For longest-term memory, as investigated in Experiment 1: Tests 2, 3a, and 3b, the average number of recalled drawings was significantly greater than that of written words when looking at the whole group of participants (Table 1, Figure 5), and for Groups 1, 2, and 3 (Figure 7). By far, the greatest difference between the number of recalled drawings and written words was for children who have difficulty remembering words (Group 3) when they remember in the longest term (Experiment 1: Tests 2, 3a, and 3b).

The group of children who recalled the greatest and moderate numbers of written words (Groups 1 and 2) in the shortest term or immediately after encoding (Experiment 1: Test 1) recalled a significantly greater number of written words than drawings. In Experiments 2 and 4, where 45 min passed from encoding to recall, the group of adults who recalled the greatest number of written words (Group 1) also recalled significantly more

written words than drawings. Adults who recalled the greatest number of words 25 min after encoding (Group 1) in Experiment 3 also generally recalled more words than drawings, but the difference was not statistically significant.

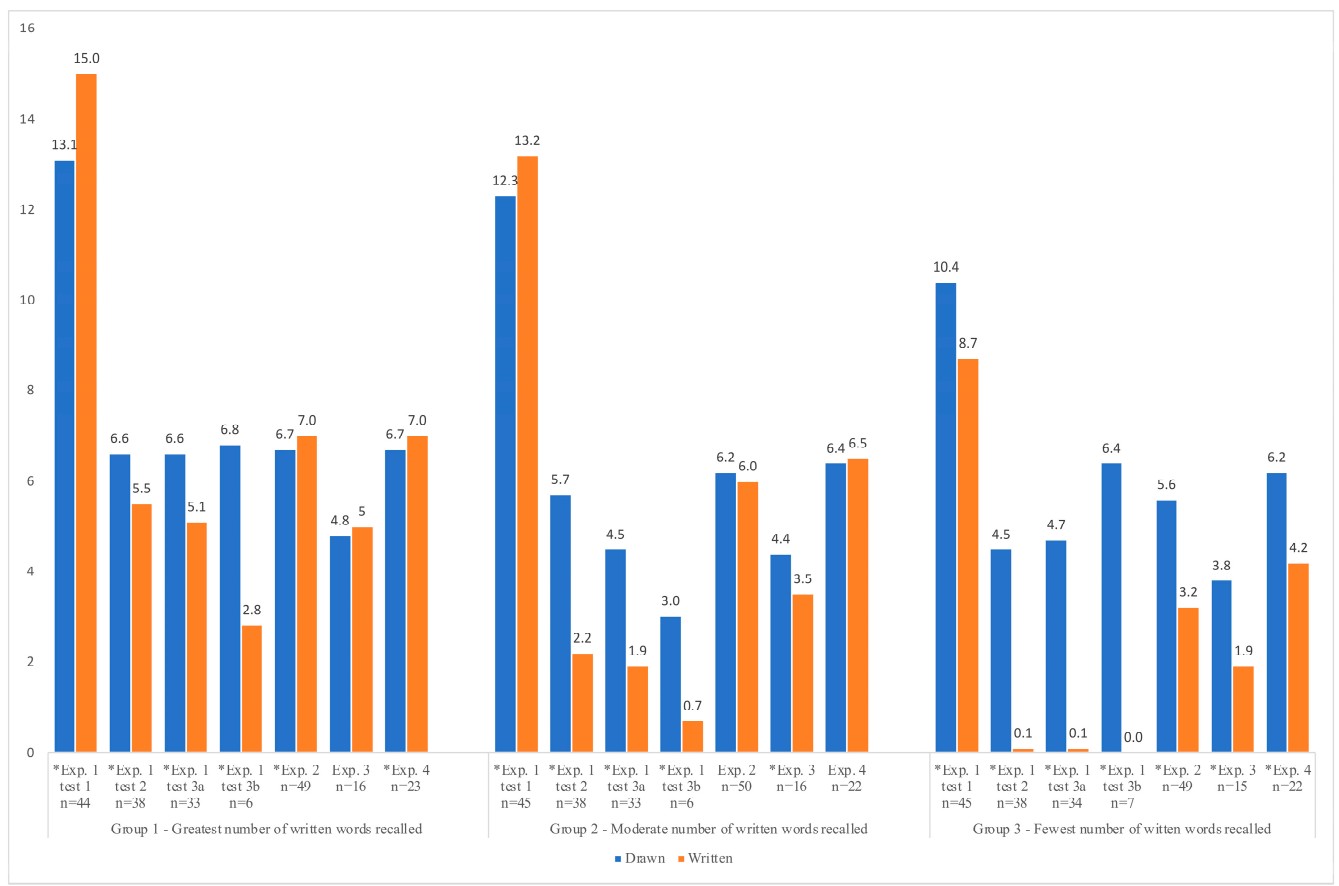

**Figure 7.** Participants in each experiment were divided into subgroups that recalled the greatest (Group 1), moderate (Group 2), and fewest (Group 3) numbers of written words. Average numbers of recalled drawings and written words for all subgroups in all experiments and tests. Asterisks (*) indicate significant differences.

One-way ANOVA tests were conducted to test differences in the effectiveness of drawing relative to writing between Groups 1, 2, and 3, sorted by the number of written words recalled. The dependent variable was the difference between the number of recalled drawn words and recalled written words (Table 2).

**Table 2.** Difference in average number of drawings recalled and average number of written words recalled by three subgroups: greatest number of written words recalled (Group 1), moderate number of written words recalled (Group 2) and fewest written words recalled (Group 3). Results from a one-way ANOVA analysis.

|  | Group 1 Greatest $(M_d - M_w)$ | Group 2 Moderate $(M_d - M_w)$ | Group 3 Fewest $(M_d - M_w)$ | *F* | *p* | *df* |
|---|---|---|---|---|---|---|
| Exp. 1 Test 1 | −1.86 | −0.84 | 1.76 | 21.97 | <0.001 | (2, 131) |
| Exp. 1 Test 2 | 1.08 | 3.53 | 4.37 | 16.54 | <0.001 | (2, 111) |
| Exp. 1 Test 3a | 1.50 | 2.55 | 4.54 | 11.38 | <0.001 | (2, 97) |
| Exp. 1 Test 3b | 3.92 | 2.33 | 6.43 | 4.05 | 0.038 | (2, 16) |
| Experiment 2 | −0.35 | 0.18 | 2.39 | 75.61 | <0.001 | (2, 145) |
| Experiment 3 | −0.25 | 0.88 | 1.93 | 16.36 | <0.001 | (2, 44) |
| Experiment 4 | −0.26 | −0.05 | 2.05 | 20.91 | <0.001 | (2, 64) |

Among adults and children with different levels of memory capacity for written words, the effectiveness of drawing for memorization was estimated by comparing the average number of recalled drawings and written words in Groups 1–3 separately. The difference between the average number of drawings and written words recalled was positive and greater in Group 3 than in Group 1 for all experiments and tests except for Experiment 1: Test 1, where the difference was negative for Groups 1 and 2. When the difference is positive, it shows that drawing is more effective than writing, but when the difference is negative, it shows that writing is more effective than drawing.

One-way ANOVA analyses revealed that the effectiveness of drawing for memorization in comparison to writing was significantly greater in Group 3 than in Group 1 in all experiments and tests. Hence, drawing in comparison to writing is especially beneficial for memorization for those who have difficulty remembering written words.

Despite the various conditions for different experiments and tests, a similar pattern was observed for most participants in Groups 1, 2, and 3 (Figure 7 and Table 2). In Group 2, which included participants who recalled a moderate number of written words, the difference between the average number of recalled written words and drawings was between the difference in the average number of recalled drawings and written words in Groups 1 and 3; this includes all cases except for Experiment 1: Test 3b, which had few participants. In addition, for shorter-term memory, for Group 1 in Experiment 1: Test 1, as well as for Group 1 in Experiments 2 and 4, the participants who easily remembered written words recalled greater number of written words than drawings. When remembering over the longest term, the children in Groups 1, 2 and 3 recalled more drawings than written words (Experiment 1: Tests 2, 3a, and 3b) (Figure 7). Moreover, by far the greatest difference between recalled drawings and written words was for children who had difficulty remembering words (Group 3).

A one-way ANOVA was also conducted to test the differences between Groups 4, 5, and 6, sorted by the number of drawings recalled, in which the dependent variable was the difference between recalled drawn words and recalled written words (Figure 8 and Table 3). Experiments 2, 3, and 4, with adults, revealed that the difference between recalled drawn words and recalled written words was not statistically different between the subgroups (Groups 4–6). All experiments among children revealed significant differences between subgroups that recalled the greatest, moderate, and fewest number of drawings. The pattern was not as clear for the comparison analysis for the subgroups, which was conducted according to the number of recalled drawings in comparison to written words, as the difference was more varied between the different subgroups when sorted according to the number of recalled drawings.

**Table 3.** Differences in the average number of drawings recalled and the average number of written words recalled by three subgroups: greatest number of drawn words recalled (Group 4), moderate number of drawn words recalled (Group 5), and fewest drawn words recalled (Group 6). Results from a one-way ANOVA analysis.

| | Group 4 Greatest $(M_d - M_w)$ | Group 5 Moderate $(M_d - M_w)$ | Group 6 Fewest $(M_d - M_w)$ | *F* | *p* | *df* |
|---|---|---|---|---|---|---|
| Exp. 1 Test 1 | 0.91 | 0.22 | −2.02 | 13.38 | <0.001 | (2, 131) |
| Exp. 1 Test 2 | 5.16 | 2.47 | 1.34 | 24.02 | <0.001 | (2, 111) |
| Exp. 1 Test 3a | 5.03 | 3.08 | 0.47 | 34.09 | <0.001 | (2, 97) |
| Exp. 1 Test 3b | 6.93 | 3.83 | 1.83 | 8.63 | 0.003 | (2, 16) |
| Experiment 2 | 0.84 | 0.96 | 0.41 | 1.05 | 0.225 | (2, 145) |
| Experiment 3 | 1.06 | 1.06 | 0.33 | 1.48 | 0.241 | (2, 44) |
| Experiment 4 | 0.91 | 0.68 | 0.09 | 1.47 | 0.237 | (2, 64) |

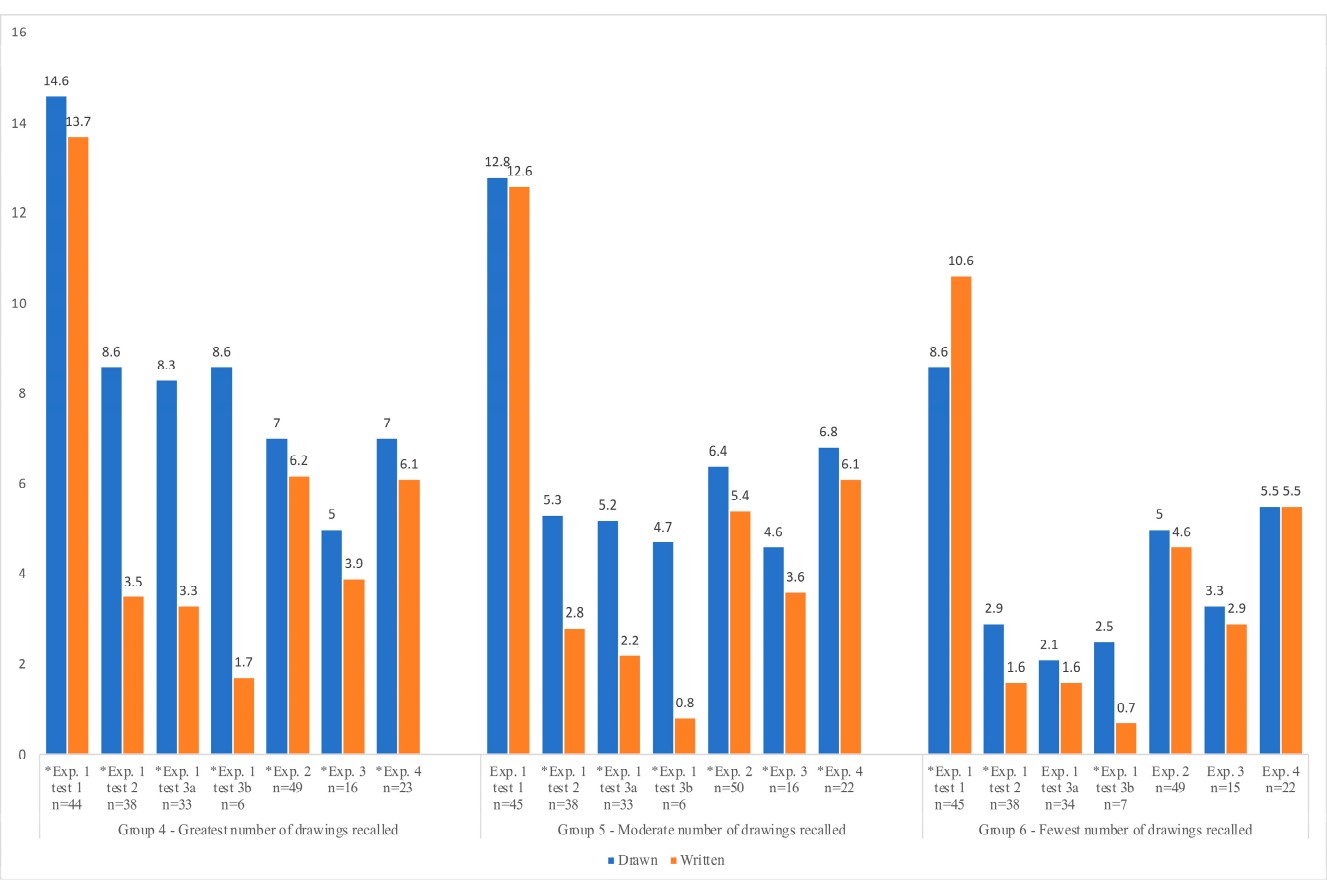

**Figure 8.** Participants in each experiment were divided into subgroups that recalled the greatest (Group 4), moderate (Group 5), and fewest (Group 6) numbers of drawings. Average numbers of recalled drawings and written words for all subgroups in all experiments and tests. Asterisks (*) indicate significant differences.

The analysis of the findings according to the subgroups that were sorted on the one hand by the number of recalled written words (Figure 7: Groups 1–3) and on the other hand by the number of recalled drawings (Figure 8: Groups 4–6) revealed somewhat different patterns. The former analysis, made for subgroups sorted according to the number of written words, showed that for all experiments and tests, the greatest difference between the number of recalled drawings and written words was in Group 3, which recalled the fewest number of words. Participants in all experiments 1–4 who recalled the greatest number of drawings (Group 4), recalled significantly more drawings than words. In all cases, for children, the difference between the number of recalled drawn words and recalled written words was greatest among those who recalled the greatest number of drawings.

## 4. Discussion

Mental images have been used to facilitate memory [16–18] and research has shown that pictures are an effective memory aid e.g., [18]. The effectiveness of drawing in comparison with writing for memorization has been studied and previously reported [1–15].

The children who participated in Experiment 1 recalled significantly more drawings than written words in the tests, where recall took place three and nine weeks after encoding. In the test in which the children recalled drawings immediately after encoding, the difference between the number of recalled drawings and written words was not significant. The finding that drawing is more effective for memorization than writing when some time has passed from encoding to recall was confirmed for adults in Experiments 2–4, which included various conditions.

*4.1. Who Benefits to the Greatest Extent from Memory Drawing?*

Previous research has shown that older adults recall proportionally more drawn words than written words compared to younger adults [4]. The participants in the present study were divided into subgroups based on the greatest, moderate, and fewest number of written words and drawings recalled to investigate whether there was a difference in the number of recalled drawings in comparison to written words for these different subgroups in all experiments and tests.

The greatest difference between the number of recalled drawings and recalled written words was found in all experiments and tests for the subgroups that recalled the fewest number of words. This shows that for people who have difficulty recalling written words, drawing is proportionally more effective than writing for memorization, compared to those who more easily remember written words. By far, the largest difference between recalled drawings and recalled written words was found when children who have difficulty remembering words memorized for the longest term or for three or nine weeks. Longest-term memory, for three and nine weeks, was not tested in adults.

*4.2. Drawing for Memory across Various Conditions*

Research has shown that the memorization benefits of drawing apply to various conditions. The benefit increases as the duration from encoding to recall is longer [7]. Drawing has also been found to be effective in different settings, and it is even more effective than writing when the time is shortened and the number of words increases [3].

Various conditions were included in the present experiments in order to study whether different scenarios would influence the way in which drawing affected memorization in comparison to writing, especially in terms of the new finding that drawing was most effective for memorization for the subgroup who recalled the fewest words. In addition to the variables presented in Ottarsdottir's [7] study, in which children participated and the timing from encoding to recall varied (Experiment 1), the present study included adult participants, different numbers of words to memorize, different durations of time for encoding and recall, various time intervals from encoding to recall, and different settings (Experiments 2–4).

When looking at the whole group of participants, the greater number of drawn versus written words was found to be statistically significant for all experiments when some time had passed from encoding to recall, regardless of the age of participants, number of words encoded, duration of time for encoding and recalling, or the research setting being a school classroom, a lecture hall, or an online webinar.

When comparing the three subgroups of participants who recalled the greatest, moderate, and fewest words, the subgroup that remembered the fewest words recalled significantly more drawings than words across all conditions tested. All subgroups who recalled the greatest, moderate, and fewest words when memorizing for the longest term (Experiment 1: Tests 2, 3a, and 3b) recalled a significantly greater number of drawings than written words across all conditions tested. The subgroup that remembered the greatest number of drawings (Group 4) recalled significantly more drawings than words across the different conditions tested, while the findings were more mixed for the other subgroups sorted according to the number of recalled drawings, especially for the subgroup that recalled the fewest number of drawings. These different findings for the groups sorted by number of drawings on the one hand and number of written words on the other hand indicate that the memory function when drawing and writing is somewhat different.

*4.3. Implications for Policy and Practice*

The findings of the present study show that both children and adults recall a significantly greater number of drawings than words when some time passes from encoding to recall. Moreover, the study revealed that children and adults who have difficulty remembering written words recall a proportionally greater number of drawings than words, relative to those who more easily remember written words.

These findings, which demonstrate the effectiveness of drawing for memorization, are essential inputs for education and art educational therapy, particularly in terms of longer-term memory, and especially for people who have difficulty remembering written words. Although this is not as important for individuals who remember written words more easily than drawings in the shorter term, both children and adults generally remember drawings better than written words in the longer term.

The effectiveness of drawing in memorizing has been claimed to be due to an integration of visual, motor, and semantic codes [3] and vivid contextual information [4]. In addition to these components, the viewpoint of the present research is that personal and emotional material can be implicit in the drawing, and therefore, the drawn content is more valuable and thus stored longer in memory than written words [7]. Additional research has found that people remember more drawings than written words, and the difference is greater for emotionally loaded words, or positive and negative words, compared to neutral words [15]. According to art therapy theories, the emotional material embedded in the drawing can be sensitive, especially for vulnerable students who have experienced difficulties and have limited support. Partly, therefore, it is claimed that the schooling situation needs to be safe enough for the students to engage in the drawing process for memorizing and learning. Understanding students' emotional lives and situations, as well as having knowledge of the emotional processes included in drawing, is claimed to be an important part of the learning process through drawing. Knowledge of the foundation of art therapy theories and methods can serve as groundwork for safely integrating drawing into therapy and education [8].

Although research has shown that doodling which is unrelated to the subject to be memorized is effective for facilitating memorization [33], another study has demonstrated that drawing related to the content of the word to be memorized is more effective for memorization than unrelated doodling [6]. Thus, it is claimed that although drawing and doodling which are unrelated to coursework are important in terms of facilitating emotional processing, coursework learning, and memorization, drawing in direct connection to the content to be memorized can still be most effective in terms of coursework learning. Therefore, in addition to doodling and spontaneous unrelated drawing for educational and emotional purposes, the following components are claimed to be important to consider when integrating drawing for coursework learning and memorization in art educational therapy and education in schools: actual drawing, emotional material that can be embedded in the drawing, and direct connection within the drawing to the coursework content to be memorized [7].

Although recent studies, that have isolated the factors of drawing and writing for memorizing certain words, have found that drawing is generally more effective for memorization than writing [1–15], the subject is more complicated in relation to actual schooling situations, as studies on the effectiveness of integrating drawing into education have shown mixed findings [34]. The aim of memory drawing is not to replace traditional coursework learning with drawing, but rather to use drawing to open up the student's ability to learn in more traditional ways through methods such as reading and writing. This can take place in various ways, such as drawing the content of a certain word when memorizing the spelling and drawing images of the content of foreign words in order to memorize their translations [7]. Other methods involve, for example, drawing components of geography for comprehension and memorizing, or drawing the content of poetry to be learned by heart. The symbolic emotional content of the memory drawing, seen from an art therapy viewpoint, is an important component of the memory drawing method [8].

Research findings on drawing for memory are also important to consider when integrating drawing into education, such as that drawing may facilitate memorization more than other, more traditional learning methods, in terms of one task but not another, as shown in Jonker et al.'s [35] research findings, whereby drawing worked well for remembering individual items but reading silently appeared to be more effective for remembering sequence within a list. Roberts and Wammes [9] also found that drawing is slightly more

effective when memorizing concrete words than abstract words. In addition, the effect of drawing on memorization is greater as the duration between encoding and recalling increases [7]. Furthermore, as shown by the findings of the present study, there are individual differences in how well the memory drawing method works, as it is more effective for individuals who have difficulty remembering their written words compared to those who more easily remember words through writing.

### 4.4. Study Limitations and Future Research

The selection of participants for Experiments 2–4 took place by including findings from individuals who attended lectures and webinars and chose to contribute their memory drawing exercises to the research. The children in Experiment 1 were randomly selected, while the adults in Experiments 2–4 might have been more akin to convenience samples. In Experiment 4, the online webinars did not allow for a high level of control because the researcher was physically absent. Participants in the webinars were free to submit their data, which they could have altered if they so desired. It is a weakness of Experiment 4 that specific measures were not implemented to ensure the reliability of data collection during the webinars. However, the findings of Experiment 4 provided by the memory drawing exercise, within the online webinars, mostly mirror the findings for Experiments 2 and 3, which indicates that the findings were not altered to a significant degree due to the absence of the researcher.

The sampling and setting for Experiments 2–4 were unusual for such research, as they took place in lectures and webinars rather than in a controlled research setting. Nevertheless, the findings of Experiments 2–4 mirror those of Experiment 1, especially in terms of those who have difficulty remembering written words. A strength of the study is that the sample was large and consisted of 134 children and 262 adults, which allowed for dividing the sample into subgroups that recalled the greatest, moderate, and fewest number of words and drawings, as well as perform separate statistical analyses for every subgroup for all experiments and tests.

A result drawn from the study is that drawing significantly facilitates longer-term memory more than writing words down and this applies to both children and adults. The new finding introduced in this article is that drawing, in comparison to writing, is especially effective for children and adults who have difficulty remembering written words, both in the short and longer terms. This is the first time that these findings have been introduced, and it reveals that the findings from Experiments 2 to 4 support the findings from Experiment 1, regardless of the variety of conditions in terms of the number of words encoded, different ages, different settings, and different durations of time devoted to encoding, as well as various durations of time between encoding and recall. The new findings provided in Experiments 2–4 provide reason to conduct further research in this area in a more controlled research setting and with a more rigorous sample selection procedure, which could increase the reliability of the findings.

Further research with fewer changes in variables between experiments could provide more accurate findings of the main factors that influence memorizing through drawing and writing in terms of different ages, various numbers of words encoded, different durations of time devoted to encoding, various durations of time between encoding and recall, and different settings. It would then be possible to separate the variables and analyze the number of recalled words and drawings for different age groups, different durations of time, and different settings to identify the main influential factors for memorization of words and drawings. Additional analysis could also be made in terms of differences between memorization of drawings and written words for certain subgroups, such as females and males, different languages, and different cultures, to observe whether drawing in comparison to writing is especially beneficial for memorization for any specific groups other than the group of individuals who have difficulty remembering written words.

## 5. Conclusions

According to the study's findings, drawing significantly aids longer-term memory more than writing down words, and this applies to both children and adults. An important new finding is that drawing, in comparison to writing down words, is most effective for children and adults who have difficulty remembering words, both in terms of short- and long-term memory and across the various conditions tested.

**Funding:** This work was funded by the Icelandic Centre for Research (RANNIS) (191043-0011/2019, 201182-0011/2020, 211410-0011/2021) and the Association of Icelandic Non-fiction Writers (2019/2021).

**Institutional Review Board Statement:** The research with the children was reviewed and approved by the Icelandic Data Protection Authority (3 May 1999 no. 99050205). Adults who participated in the experiments provided written informed consent for participating in the study and for their data to be used for research purposes, which is the present ethical requirement requested according to the regulations of the Icelandic Data Protection.

**Informed Consent Statement:** Informed consent was obtained from all subjects involved in the study.

**Data Availability Statement:** Data supporting reported results can be found at https://dataverse.harvard.edu/dataset.xhtml?persistentId=doi:10.7910/DVN/UXAOYA (accessed on 27 April 2024).

**Acknowledgments:** The author would like to thank all participants in the research for contributing their time and effort when completing the memory drawing research and for sharing their valuable data. I also would like to thank my son, Jón Karl Sigurðsson mathematician, for reading the manuscript and providing feedback. I would also like to thank my father, Óttar Yngvason, Attorney of Law, for his encouragement, support, and discussions regarding the research project.

**Conflicts of Interest:** The author declares no conflict of interest.

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
