# Peer review of "Experiments on the Efficacy of Drawing for Memorization among Adults and Children with Varying Written Word Memory Capacities: A Two-Way Crossover Design"

_education, doi:10.3390/educsci14050470_

Round 1

Reviewer 1 Report

Comments and Suggestions for Authors

Dear Authors

Thank you for submitting your manuscript to the Journal of Education Sciences.

Your manuscript addresses the topic of memorization among children and adults through drawing.

Content-wise, the manuscript is based on: 0) an abstract; 1) an introduction outlining the state-of-the-art and motivating the relevance of your study; 2) description of research design, methods used, and participants; 3) a detailed description of the four experiments (7 tests in total?); 4) results of the experiments and statistical analysis using among others ANOVA tests; 5) a discussion and conclusion highlighting the value of drawing for memorization in comparison to written words, as well as arguing for the relevance in learning situations and in general when working with children who have experienced stress and/or trauma.

Your overall idea to focus on drawing for memorization (and learning) is highly relevant and important in the debate about meeting human diversity and establishing better learning environments in general. For most parts the structure of the manuscript is clear and well-written, and I think there is a need for more of this kind of exploratory work elaborating on developing and testing insights on drawing. Hence, the newsworthy of the topic is high, and I strongly support your intent and ambition of the manuscript.

Nevertheless, the conclusions of the manuscript, as well as the discussion and critical reflection of the studies are less clear to me as an external reader and scholarly reviewer. At present I miss details about the participants to follow the parts 696-706 + 762-792. Here you mention “..traditional ways of learning can become easier by integrating drawing…”, “..to offer understanding and empathy, and possibly to refer to students who need deeper emotional work to therapy…awareness to the emotional factors…”…for wellbeing…” and again: “…it is possible that the ability to memorize through drawing, in comparison to writing, is less disturbed by experiences and conditions, such as stress and/or trauma…”…you also mention with reference to other scholars patients with Alzheimer’s disease, cognitive impairments and older people, as well as children with trauma. But it is not clear to me if your specific participants in any of the conducted experiments belongs to these groups? So, I am in doubt of the strength of the ‘evidence’ so to speak of your studies in relation to this discussion.

You mention in line 764, that this is subject for further research, but still, the above statements seem to occupy quite a large part of your reflection. And there is no direct mentioning of how much or how little your current research studies are cable of informing you in these matters. At present, I do not know if any of the participants have suffered from stress and/or trauma… or how their emotional state-of-being is before the experiments. Consequently, the final take-home-message of the manuscript is relatively blurred.

It would improve the readability a great deal, towards the end, if a clearer outline of the participants was added before part 4, or you try to limit the critical reflections concerning wellbeing and emotional state-of-being to what your actual research studies are able to clarify.   

If you succeed in addressing the above points in your revised version, I think you have a very promising paper with an important ‘take-home-message’, that would be interesting to read.

Thank you.

Author Response

Thank you for carefully reading the manuscript and providing a helpful review. Your general positive feedback about the manuscript is encouraging and motivating. I appreciate your valuable comments and suggestions for improvement. I have revised the manuscript based on your suggestions.

“Nevertheless, the conclusions of the manuscript, as well as the discussion and critical reflection of the studies are less clear to me as an external reader and scholarly reviewer.

At present I miss details about the participants to follow the parts 696-706 + 762-792. Here you mention “..traditional ways of learning can become easier by integrating drawing…”, “..to offer understanding and empathy, and possibly to refer to students who need deeper emotional work to therapy…awareness to the emotional factors…”…for wellbeing…” and again: “…it is possible that the ability to memorize through drawing, in comparison to writing, is less disturbed by experiences and conditions, such as stress and/or trauma…”…you also mention with reference to other scholars patients with Alzheimer’s disease, cognitive impairments and older people, as well as children with trauma. But it is not clear to me if your specific participants in any of the conducted experiments belongs to these groups? So, I am in doubt of the strength of the ‘evidence’ so to speak of your studies in relation to this discussion.

You mention in line 764, that this is subject for further research, but still, the above statements seem to occupy quite a large part of your reflection. And there is no direct mentioning of how much or how little your current research studies are cable of informing you in these matters. At present, I do not know if any of the participants have suffered from stress and/or trauma… or how their emotional state-of-being is before the experiments. Consequently, the final take-home-message of the manuscript is relatively blurred.

It would improve the readability a great deal, towards the end, if a clearer outline of the participants was added before part 4, or you try to limit the critical reflections concerning wellbeing and emotional state-of-being to what your actual research studies are able to clarify.” 

Thank you for the comments. It is correct that there is no information within the manuscript about the participants’ emotional states regarding whether they suffered from stress and/or trauma. Therefore the discussion is speculative, as you point out. I did not have any information about this at the time of the research or at any point later.

Therefore, I have followed your latter suggestion to meet the comment. I have deleted the following paragraphs, concerning the wellbeing and emotional state-of-being of the participants, from the revised manuscript.

  1. 241-259
  2. 241 Revised manuscript

Research has also shown that patients with Alzheimer’s disease and people with mild cognitive impairment remember pictures as well as other healthy older people, which indicates that the ability to memorize pictures may be intact in these populations [36–39]. Ally et al. [37] showed that the picture superiority effect was greater for older adults than for younger adults and suggested that pictures may compensate for the impaired memory processes in older adults.

Similarities in brain function between patients with Alzheimer’s disease and those with post-traumatic stress disorder (PTSD) have been reported. One similarity in both populations is a reduction in hippocampal volume [40]. If there are some similarities in the brain functions for memory in people with PTSD and Alzheimer’s, which may be partly related to functions in the hippocampus, and if people with Alzheimer’s remember pictures as well as healthy older people, this may support the hypothesis that drawn pictures are an effective integration for memorizing in educational and therapeutic contexts, especially for individuals who have experienced trauma. Drawing might thus add an important memory pathway for people who have experienced stress and/or trauma, as in the case of children who participated in the AET case study [7]. The question of whether people who have experienced trauma remember pictures and drawings more easily than words in comparison to people who have not experienced trauma remains a subject for further research.  

  1. 703-711
  2. 759 Revised manuscript

In other words, traditional ways of learning can become easier by integrating drawing into coursework learning to a suitable extent under safe enough circumstances. Although awareness of the emotional factors is important when students draw, the aim is not that the educator provides therapy for the students but rather to offer understanding and empathy, and possibly to refer students who need deeper emotional work to therapy. Awareness of the emotional factors involved, along with connecting the drawing to some degree to coursework learning material, are seen as important parts of integrating drawing into education in schools, both for well-being and coursework learning. 

  1. 763-797
  2. 835 Revised manuscript

Different patterns for the number of recalled drawings and written words were found for the subgroups of participants who recalled the greatest, moderate, and fewest written words on the one hand, and the greatest, moderate, and fewest drawings on the other, which indicates that the memory processes for these two memorizing methods are somewhat different (Tables 2 and 3; Figures 7 and 8). It is possible that the ability to memorize through drawing, in comparison to writing, is less disturbed by experiences and conditions, such as stress and/or trauma, dyslexia, or age, which is a subject for further research.

Patients with Alzheimer’s disease and people with mild cognitive impairment remember pictures as well as other healthy older people [36–39] and the picture superiority effect is greater for older adults than for younger adults [37]. These findings also point to further research inquiries about whether people who have experienced trauma, such as the children who took part in the qualitative AET case study [7], remember pictures and drawings more easily than words. Tran [15] found that negative and positive drawn content of written words is more easily memorized than neutral words, which indicates that emotions may play a role in the effectiveness of memorizing through drawing. From an art therapeutic perspective, drawing is more effective than writing for memorization partly because of the implicit personal emotional content in drawing. However, the question remaining is whether people who are dealing with complex emotional issues due to experiencing stress and/or trauma are likely to be in the group that has the most difficulty remembering written words. This may be especially the case for those who deal with emotional difficulties that are partly related to not remembering the trauma, and it is possible that these individuals have also been unable to orally express their stressful and/or difficult experiences and emotions, which was often the case with the children who participated in the AET case study.

The greatest difference between memorization of written words and memorization of drawn content of words was found for children who have difficulty remembering words when they memorize the longest term; this group may include children who have experienced stress and/or trauma and are dealing with specific learning difficulties. This is a research inquiry that may help to target and further understand the process of memory drawing within education and art educational therapy, especially for individuals who have experienced stress and/or trauma and deal with specific learning difficulties. Further research into drawing for this population would be to investigate whether drawing is more effective for remembering forgotten traumatic events than talking about the experiences or expressing them in writing.

I wish to thank the reviewer again for the helpful and insightful comments on the manuscript.

Reviewer 2 Report

Comments and Suggestions for Authors

This manuscript presents a series of experiments investigating the effectiveness of drawing versus writing for memorization in both children and adults, particularly focusing on individuals with specific learning difficulties and those who have experienced stress or trauma. The research design incorporates qualitative case studies and quantitative experiments, with ethical approval obtained for the studies involving children. The experiments involve various conditions such as different age groups, numbers of words encoded, durations of time devoted to encoding, durations of time between encoding and recalling, and different settings.

The manuscript provides clear descriptions of the research background, design, procedures, and participants involved in each experiment. The methodology appears robust, utilizing a two-way crossover design for the experiments with children and incorporating diverse settings for experiments with adults. The inclusion of multiple experiments with varying conditions enhances the comprehensiveness of the study.

Overall, the manuscript presents an intriguing exploration of the potential benefits of integrating art therapy techniques into educational settings for improving memory retention, particularly for individuals with specific learning difficulties or those who have experienced stress or trauma.

Questions:

  1. Could you provide more details about the specific methods used to ensure the ethical conduct of the research, particularly concerning the involvement of children in the experiments?

  2.  
  3. How were the participants recruited for the experiments conducted with adults? Were there any specific criteria for participation?

  4.  
  5. In Experiment 1, could you elaborate on the rationale behind choosing a three-week and nine-week recall period? How were the participants reminded of the encoding experience during the nine-week recall period?

  6.  
  7. For Experiment 4, what measures were implemented to ensure the reliability of data collection during the online webinars, considering the absence of the researcher during the memory drawing test?

  8.  
  9. Can you provide insights into the practical implications of the findings for educational and therapeutic interventions, especially in terms of integrating drawing techniques for enhancing memory retention?

Author Response

Thank you for your helpful questions and comments on the manuscript. Your general positive feedback is encouraging and motivating. Below, I have provided explanations and stated the implemented revisions I made throughout the paper according to the questions you posed. These are listed below, point by point, in reference to each question.

  1. Could you provide more details about the specific methods used to ensure the ethical conduct of the research, particularly concerning the involvement of children in the experiments?
  2. 307-311
  3. 328-334 Revised manuscript

“The quantitative research conducted on the children by the researcher using a two-way crossover design was reviewed and approved by the Icelandic Data Protection Authority. Adults who participated in the experiments provided written informed consent for their data to be used for research purposes, which is an ethical requirement requested according to the regulations of the Icelandic Data Protection Authority.

The Icelandic Data Protection Authority (no. 99050205) provided ethical approval for the research with the children. The research ensured ethical conduct, as the children were free to participate in the research and no sensitive personal information was collected. Also, no personal information is identifiable within the research or in its findings.

Kindly note that at the time of the research, there were different views, from now, on children participating in research, and formal consent from them was not necessary, common, or needed as no sensitive personal information was collected. Also, because there is no identifiable personal information included in the memory drawing test/exercise, there was no requirement to collect consent from the parents.

Adults who participated in the experiments provided informed consent for their data to be used for research purposes, which is an ethical requirement according to the regulations of the Icelandic Data Protection Authority. 

In order to meet this comment, I have added this sentence to the manuscript:

  1. 309
  2. 330-331 Revised manuscript

“In order to ensure the ethical conduct of the research, the children were free to participate in the research, and no sensitive personal information was collected“

  1. How were the participants recruited for the experiments conducted with adults? Were there any specific criteria for participation?

See information about how the participants were recruited for the experiments conducted with adults.

  1. 357-359 and L. 381-383.
  2. 383-385 and 407-409 Revised manuscript

Experiments 2-4 were integrated into lectures given by the researcher on memory drawing research, where the participants completed a memory drawing exercise similar to that in Experiment 1. … On the fourth page, the participants were invited to submit the findings of their memory drawing exercises and give consent for the anonymous use of the data for research purposes.

  1. 726-728
  2. 774-776 Revised manuscript

The selection of participants for Experiments 2–4 took place by including findings from individuals who attended lectures and webinars and chose to contribute their memory drawing exercises to the research.

The criteria for selecting participants were embedded in the nature of the setting where the research took place, where all participants had chosen to attend the lecture, and therefore were probably interested in taking part in the memory drawing exercise. The participants were, however, free to participate or not in the exercise as they wished. It would be possible to conduct further research with different populations, such as people from different backgrounds with different interests, e.g. individuals who work in companies who do not have anything to do with art or education. It would be interesting to compare the results for these different populations in further research.

In order to further clarify, I have added this text:

Experiments 2-4 were integrated into lectures given by the researcher on memory drawing research, where the audience was invited to take part in the memory drawing research.

  1. In Experiment 1, could you elaborate on the rationale behind choosing a three-week and nine-week recall period?

The following reasons are given in the manuscript for choosing to investigate how much the children remembered in the longer term.

  1. 332-363
  2. 358-331 Revised manuscript

The results of Test 1 did not show a difference in the median number of recalled written words and drawings immediately after encoding. Drawing may relate to personal conscious and unconscious meanings and memories, which may facilitate storage at a deeper level, which in turn might cause the drawing to be stored in memory longer than writing. These speculations resulted in the decision to investigate again, in Tests 2, 3a, and 3b, how much the children recalled after a certain time.

Three weeks were chosen because it provided sufficient time for the content to transition from the shorter-term memory to the longer-term memory. I chose nine weeks to explore a still longer-term memory, as it was likely that some of the content would disappear from the long-term memory. The manuscript could include this explanation, but I’m not sure if it is needed.

  1. How were the participants reminded of the encoding experience during the nine-week recall period?

The participants were reminded when completing Test 2 three weeks after the original encoding. The manuscript states:

  1. 338-345
  2. 364-367 Revised manuscript

All participants in Experiment 1: Tests 2, 3a, and 3b participated in Test 1. For Test 2, 114 of the 134 children recalled three weeks after encoding in Test 1. For Test 3a, 100 of the 134 children were recalled nine weeks after the original encoding, having undergone a recalling and reminding process six weeks earlier.

In order to clarify, I have added to the sentence that the reminder took place when the participants completed Test 2:

All participants in Experiment 1: Tests 2, 3a, and 3b participated in Test 1. For Test 2, 114 of the 134 children recalled three weeks after encoding in Test 1. For Test 3a, 100 of the 134 children were recalled nine weeks after the original encoding, having undergone a recalling and reminding process six weeks earlier when completing Test 2.

  1. For Experiment 4, what measures were implemented to ensure the reliability of data collection during the online webinars, considering the absence of the researcher during the memory drawing test?

In order to further state the weakness embedded in Experiment 4, I have added this text about the subject (L. 779-785 Revised manuscript):

In Experiment 4, the online webinars did not allow for a high level of control because the researcher was physically absent. Participants in the webinars were free to submit their data, which they could have altered if they so desired. It is a weakness of Experiment 4 that specific measures were not implemented to ensure the reliability of data collection during the webinars. However, the findings of Experiment 4 provided by the memory drawing exercise, within the online webinars, mostly mirror the findings for Experiments 2 and 3, which indicates that the findings were not altered to a significant degree due to the absence of the researcher.

  1. Can you provide insights into the practical implications of the findings for educational and therapeutic interventions, especially in terms of integrating drawing techniques for enhancing memory retention?

Yes, thank you for this question. Although this subject is not the main focus of the manuscript, I think it is an improvement to add some information about how to integrate memory drawing for enhancing memory retention. Therefore, I have added the following explanations, referencing further information about how to integrate memory drawing into education and therapy.

  1. 753-759 Revised manuscript

This can take place in various ways, such as drawing the content of a certain word when memorizing the spelling and drawing images of the content of foreign words in order to memorize their translations [7]. Other methods involve, for example, drawing components of geography for comprehension and memorizing, or drawing the content of poetry to be learned by heart. The symbolic emotional content of the memory drawing, seen from an art therapy viewpoint, is an important component of the memory drawing method [8]. 

I wish to thank the reviewer again for the helpful comments on the manuscript.
